# Comparison of the Grain Quality and Starch Physicochemical Properties between *Japonica* Rice Cultivars with Different Contents of Amylose, as Affected by Nitrogen Fertilization

Yajie Hu 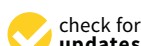, Shumin Cong and Hongcheng Zhang *

Jiangsu Co-Innovation Center for Modern Production Technology of Grain Crops, Jiangsu Key Laboratory of Crop Genetics and Physiology, Yangzhou University, Yangzhou 225009, China; huyajie@yzu.edu.cn (Y.H.); cong202105@163.com (S.C.)
* Correspondence: hczhang@yzu.edu.cn

**Abstract:** In order to determine the effects of nitrogen fertilizer on the grain quality and starch physicochemical properties of *japonica* rice cultivars with different contents of amylose, normal amylose content (NAC) and low amylose content (LAC) cultivars were grown in a field, with or without nitrogen fertilizer (WN). The relationships between the amylose content, starch physicochemical properties and eating quality were also examined. Compared with WN, nitrogen fertilizer (NF) significantly increased the grain yield but markedly decreased the grain weight. In addition, the processing quality tended to improve, but the appearance quality and eating quality deteriorated under NF application. The grain yield was similar between NAC and LAC cultivars. However, the grain quality and starch physicochemical properties were significantly different between NAC and LAC cultivars. The palatability of the cooked rice was significantly higher in the LAC than in NAC cultivar, which was due to its lower amylose content, protein content, hardness, and retrogradation enthalpy and degree, and its higher stickiness, peak viscosity, breakdown, relative crystallinity and peak intensity. The amylose content and protein content were significantly negatively correlated with the palatability. The amylose content was significantly positively correlated with the final viscosity and setback, and was significantly negatively correlated with the relative crystallinity, peak intensity, gelatinization enthalpy and breakdown. Palatability was significantly positively correlated with peak viscosity, breakdown and peak intensity, and was significantly negatively correlated with the final viscosity, setback, and retrogradation enthalpy and degree. Therefore, the selection of a low amylose content *japonica* rice cultivar grown without nitrogen fertilizer can reduce the amylose and protein contents, as well as improving the pasting properties, starch retrogradation properties and eating quality of the cooked rice.

**Keywords:** nitrogen fertilizer; *japonica* rice cultivars; grain quality; starch physicochemical properties



## 1. Introduction

Rice (*Oryza sativa* L.) is one of the most staple food grains for more than half of the world's people, especially in China, where rice provides nutrients for two-thirds of the population [1]. With the increase in the population and the improvement of living standards in China, the demand for a high-yield, high-quality rice grain is rapidly increasing. Recently, the structural reform of the cereal grain supply was proposed because of the relative surplus of national cereal grain in China [2,3]. As a result, the focus has shifted from high yield to high quality in rice breeding and cultivation. In order to improve the quality of *japonica* rice, some cultivars with a high eating quality and low amylose content (AC: 10% to 15%) have been released in China, such as Nangeng 46, Nangeng 5055, Nangeng 9108, Suxianggeng 100, Zaoxianggeng 1, Huruan 1212 and Songxianggeng 1018, which have the characteristics of being soft but not sticky after cooking, and are defined as soft *japonica* rice [4,5].

Rice quality is a very important factor affecting the choice of producers and consumers, including milling, appearance, nutrition and eating quality [6]. The four main quality traits are related to the content and composition of starch and protein in rice endosperm. The appearance quality, indicated by the opaqueness or chalkiness, is affected by the shape, size, and packing of starch granules, and by the starch composition [6]. The protein content stored in the seed determines the nutritional quality [7]. Of the four quality traits, eating quality is the most important, which is affected by multiple factors related to the amylose content, amylopectin content, gel consistency, gelatinization temperature and protein content [8]. A low amylose content is the main factor determining the high taste quality of cooked rice [9]. The protein content is also involved in the cooking quality, by impeding starch gelatinization (Chen et al. 2012; Ma et al. 2017). The pasting properties reflect the variation in the viscosity of the starch when rice is cooked, and in previous studies, a high taste quality was associated with a high peak viscosity and breakdown, and a low setback [10,11].

Starch is the dominant component in rice grains, accounting for approximately 90% of the endosperm's weight, and it is essentially composed of linear amylose and moderately branched amylopectin [12]. Previous studies indicate that the starch composition and structure influence the starch functional properties [13,14]. Cai et al. [15] found that the amylose and amylopectin contents, as well as the crystalline structure (relative crystallinity, lamellar peak intensity and lamellar distance), determine different functional properties; starch with a higher amylose content always shows faster retrogradation, leading to the higher setback viscosity of the starch. Wang et al. [16] found that rice starch with a lower amylose content had a higher relative crystallinity and short-range molecular order, which contributed to its higher swelling power, peak viscosity and breakdown viscosity. However, little information is available on the correlations between the starch structure and physicochemical properties and the rice grain quality, especially the eating quality.

The difference in the grain quality and starch physicochemical properties of rice cultivars with normal amylose content (NAC) and low amylose content (LAC) has been reported previously [17,18]. Nevertheless, the amylose content of rice cultivars with low AC is generally more susceptible to a varying environment than those with high AC. Cheng et al. [17] suggested that the effects of high temperatures on the AC of rice are cultivar-dependent, with high temperatures reducing the AC in a low-AC cultivar but increasing the AC in a high-AC cultivar.

Nitrogen (N) is a crucial factor that affects the AC, grain quality and the starch physicochemical properties. In previous studies, the amylose content decreased with an increase in the N level in NAC [19] and LAC [20] rice cultivars. How, the way in which the susceptibility of AC to N deficiency is affected by NAC and LAC cultivars is unclear. Several studies reported that N application improved the processing quality; in particular, topdressing N fertilizers increased the head rice percentage [21,22]. Zhu et al. [20] reported that increased N application decreased the appearance quality and improved the nutritional quality. Gao et al. [19] suggested that rice's taste value increases with a decrease in the N level, with the maximum value reached under an N deficiency. Therefore, we hypothesized that the rice eating quality would be improved by selecting an LAC rice cultivar grown under an N deficiency.

The objectives of this study were: (1) to compare the grain quality and starch physicochemical properties between NAC and LAC *japonica* rice cultivars with or without N fertilizer, and (2) to determine the relationships between the rice eating quality and the amylose content, protein content and starch properties. The traits of the rice quality (milling, appearance, nutrition, and eating quality), starch structure (starch granule size, relative crystallinity and lamellar structure) and physicochemical properties (pasting and gelatinization properties) were determined in order to understand their relationships.

## 2. Materials and Methods

### 2.1. Plant Materials and Growth Conditions

Field experiments were conducted in a paddy field in Shengao Town, Jiangyan County, Jiangsu Province, China (120.13° E, 32.58° N) in 2016 and 2017. In the paddy field, the clay soil had the following nutrient contents in 2016 and 2017, respectively: 27.1 and 26.9 g kg$^{-1}$ organic matter; 1.8 and 1.7 g kg$^{-1}$ total N; 18.9 and 18.3 g kg$^{-1}$ Olsen-P; and 95.2 and 95.9 mg kg$^{-1}$ available K. The soil nutrients were determined by the standard procedures [23]. The precipitation, sunshine hours and temperature during the rice growing season across the two years are shown in Figure 1.

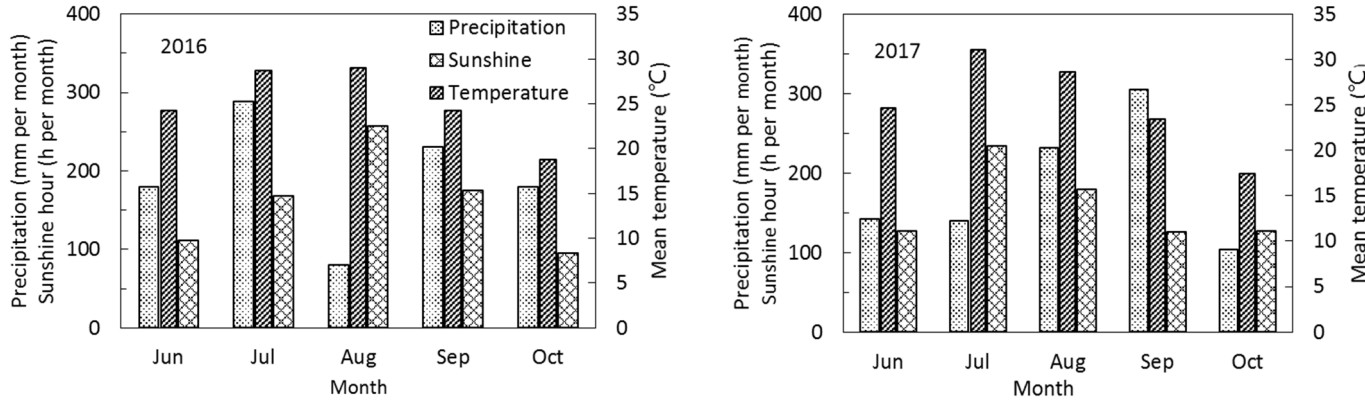

**Figure 1.** Precipitation, sunshine hours and temperature during the rice growing season of 2016 and 2017 in Jiangyan, Southeast China. The precipitation and sunshine hours are monthly totals. The temperatures are the monthly averages.

Eight conventional *japonica* rice cultivars or lines with different amylose contents were investigated in this study. We included four NAC cultivars with amylose contents ranging from 15% to 17.8% and four LAC cultivars with amylose contents ranging from 9.2% to 14.2%. The specific information on the rice cultivars or lines is shown in Table 1.

**Table 1.** The information on the *japonica* rice cultivars or lines with different contents of amylose.

| Cultivar Type | Cultivar/Line | Amylose Content of Official Release (%) | Year of Official Release | Cross Information | Breeding Organization |
|---|---|---|---|---|---|
| NAC | Jingeng 818 | 17.8 | 2014 | Jindao 9618 × Jindao 1007 | Tianjing Academy of Agricultural Science |
| | Sugeng 815 | 15.0 | 2014 | Zhengdao 99 × Wuyungeng 11 × Yandao 1229 | Jiangsu Zhongjiang Seed Industry Company |
| | Liangeng 7 | 16.2 | 2014 | Zhendao 88/Zhonggeng 8415 × Zhongjingchuan 2/Wuyugeng 3 | Lianyungang Academy of Agricultural Science |
| | Wuyungeng 27 | 17.2 | 2012 | Jia 45/9520 × Wuyungeng 21 | Wujing Rice Research Institute |
| LAC | Songzaoxiang 1 | 13.2 | 2014 | Zaoxiangruanfan 2 × Zaoxiangchangligeng | Shanghai Songjiang District Agricultural Technology Extension Center |
| | Zaoxianggeng 1 | 9.2 | 2019 | Nanjing 46 × Wuyugeng 21 | Changshu Agricultural Research Institute |
| | Yanggeng 239 | 12.9 * | Not released | / | Yangzhou Academy of Agricultural Science |
| | Ning 4725 | 14.2 * | Not released | / | Nanjing Agricultural University |

NAC, *japonica* rice with normal amylose content; LAC, *japonica* rice with low amylose content. * The amylose content was determined in this experiment.

*2.2. Treatments*

The experiment was conducted in a rice–wheat rotation and was a split-plot design, with the N rate as the main plot and the cultivar amylose content as the split plot. The two N levels were 'no N application' (WN, 0 kg ha$^{-1}$) and 'N fertilizer' (NF, 225 kg ha$^{-1}$). The seeds were sown on 13 June 2016 and 15 June 2017. The seeding rate was 60 kg ha$^{-1}$. The size of each plot was 12 m$^2$ (4 m × 3 m). Three replicates were included in the experiment. Phosphorus (80 kg ha$^{-1}$ as single superphosphate) and potassium (120 kg ha$^{-1}$ as potassium chloride) were applied and incorporated before the direct sowing in both nitrogen treatments. In the NF treatment, basal N (90 kg ha$^{-1}$ as urea) was applied and incorporated before the direct sowing, and top dressing N was also applied at the tillering (90 kg ha$^{-1}$ as urea) and panicle branch differentiation (45 kg ha$^{-1}$ as urea). Water, weeds, insects and diseases were controlled as required to avoid yield loss.

*2.3. Grain Quality and Starch Property Measures*

Once the rice had matured, the grain yield was determined from a harvest area of 6.0 m$^2$ in the middle of each plot. The yield components, i.e., the number of panicles per square meter, the number of spikelets per panicle, the percentage of filled grains, and the grain weight, were determined. The harvested grains were air-dried to 14% grain moisture content and then stored at 4 °C for three months. The rice quality analyses were performed according to GB/T 17891-2017.

### 2.3.1. Processing Quality

The rice grains were passed through a dehusker to obtain brown rice. The brown rice was polished to obtain milled rice. In order to obtain head-milled rice, grain with a length longer than 3/4 of its total length was separated from the milled rice. The brown rice, milled rice, and head-milled rice are expressed as percentages of the total grain weight.

### 2.3.2. Appearance Quality

The chalkiness was evaluated on 100 milled grains per plot. Chalky kernel rate (%) = the number of chalky kernels/100 milled grains × 100. Chalky area (%) = the chalky area/the total area of the kernel. Chalkiness degree (%) = the chalky kernel rate × the chalky area.

### 2.3.3. Nutritional Quality

The protein content and amylose content were measured in a 30 g sample of milled rice with a grain analyzer (Infratec 1241, Foss, Denmark). The protein content indicates the nutritional quality of rice. The amylose content is related to the eating quality.

### 2.3.4. Eating Quality

The sensory properties (palatability, hardness, stickiness) of the cooked rice were measured using an STA-1A rice sensory analyzer (Satake, Japan), which reflects the eating quality. Milled rice, 30 g, was washed in a stainless steel container and then transferred into a 50 mL aluminum box containing 40 mL water. The milled rice was cooked in an electric rice cooker (Z06YA3-S2, Supor, China). After the cooking, the sensory properties of the cooked rice were determined.

### 2.3.5. Pasting Properties

The milled rice was ground into flour. The rice flour pasting properties were determined using a rapid viscosity analyzer (RVA, Super 3, Newport Scientific, Australia). A total of 3 g flour sifted through 0.15-mm sieves was mixed with 25 g pure water in the RVA sample can. The peak viscosity, trough viscosity, final viscosity in cP (centipoise) and their derivative parameters, i.e., breakdown (peak viscosity-trough viscosity), setback (final viscosity-peak viscosity), and consistency (final viscosity-trough viscosity) were recorded with matching Software of Thermal Cline for Windows (TCW).

### 2.3.6. Flour and Starch Isolation

Polished rice was ground into flour in a mill (FOSS 1093 Cyclotec Sample Mill, Sweden) in order to pass through a 0.5 mm screen. The starch was isolated from the rice grains according to the method of Zhang et al. [13]. Polished rice grains (10 g) were soaked in an NaOH solution (0.2%) for 24 h. After soaking at room temperature, the swollen grains were rinsed and homogenized in a blender (IKA-T RCT-Basic, Germany) at 1500 rpm for 3 min. Then, 5 mg protease K was added to the slurry and mixed for 24 h at 37 °C. The slurry was sieved through a 150 μm mesh. The residue was mixed with the original volume of the 0.2% NaOH solution and passed through a 150 μm mesh. The combined starch filtrates were centrifuged at $3000 \times g$ for 15 min, and the supernatant was decanted. The process was repeated six times until the yellow tailings were removed. The starch sample was washed with ethanol, dried at 30 °C and stored in a container at 4 °C.

### 2.3.7. X-ray Diffraction Analysis of the Starch

X-ray diffraction (XRD) patterns were obtained using an X-ray diffractometer (D8 Advance, Bruker, Germany). The starch samples were analyzed by the diffractometer at 200 mA and 40 kV, with a diffraction angle (2θ) that ranged from 3 to 35° (2θ) at a scanning speed of 0.02°. The relative crystallinity (%) was quantified as the ratio of the crystalline area to the total area.

### 2.3.8. Small-Angle X-ray Scattering Analysis of the Starch

The lamellar structure of a starch sample was determined by using a Bruker NanoStar SAXS instrument equipped with a Vantec 2000 detector and pin-hole collimation for point focus geometry according to the method of Cai et al. [15]. The variables of the small-angle X-ray scattering (SAXS) were analysed according to the simple graphical method.

### 2.3.9. Gelatinization Properties

The starch gelatinization properties were measured using differential scanning calorimetry (DSC, 200-F3, Netzsch, Germany). In total, 5mg starch was weighed in an aluminum pan and mixed with 15 μL water. The samples were stored at 4 °C for 12 h. After incubating at room temperature for 1 h, the samples were then heated from 20 °C to 130 °C at a rate of 10 °C min$^{-1}$. The samples were analyzed in triplicate.

### 2.4. Statistical Analyses

The data were analyzed using Microsoft Excel 2013 and SPSS 17.0 (SPSS, Chicago, IL, USA). Following the ANOVA, the means comparison was based on a least significant difference (LSD) test at $p < 0.05$. The data of the grain yield components were analyzed as the average of two years; only the 2016 data were used for the analysis of the grain quality and starch properties.

## 3. Results

### 3.1. Grain Yield and Yield Components

The grain yield and yield components of both types of *japonica* rice cultivars varied with the N level (Table 2). Compared with the yield (kg ha$^{-1}$) in WN, the grain yield of both types of *japonica* rice cultivars increased significantly in NF in 2016 and 2017. In the same N treatment, there were no significant differences in the grain yield between the NAC and LAC cultivars. Of the yield components, the panicles and spikelets per panicle increased significantly, and the grain weight decreased significantly in NF compared with those in WN. In both NF and WN, the yield components were similar between the NAC and LAC cultivars.

### 3.2. Milling and Appearance Quality

Compared with those in WN, the measures of the milling quality (brown rice, milled rice, and head-milled rice) and appearance quality (chalkiness and chalky grain percentage) in

both types of rice cultivars increased slightly in NF (Table 3). Therefore, with N application, the milling quality improved, but the appearance quality deteriorated in both types of rice cultivars. In the same N treatment, the percentages of milled rice and head-milled rice and the measures of the appearance quality were lower in NAC than in LAC cultivars (Table 3), whereas the brown rice rate was higher in NAC than in LAC cultivars. Therefore, the LAC cultivars had a better milled rice quality but inferior appearance quality.

**Table 2.** Yield and yield components of the *japonica* rice cultivars with different contents of amylose, grown with or without nitrogen fertilization.

| Nitrogen Treatment [1] | Cultivar Type [2] | Cultivar | Panicles ($\times 10^4$ ha$^{-1}$) | Spikelets per Panicle | Filled Grain Percentage (%) | Grain Weight (mg) | Yield in 2016 (kg·ha$^{-1}$) | Yield in 2017 (kg·ha$^{-1}$) |
|---|---|---|---|---|---|---|---|---|
| NF | NAC | Jingeng 818 | 379.80 | 88.60 | 94.89 | 27.80 | 8.46 | 8.80 |
| | | Sugeng 815 | 443.10 | 86.50 | 84.94 | 26.76 | 8.30 | 8.59 |
| | | Liangeng 7 | 427.50 | 91.70 | 93.64 | 25.33 | 9.44 | 9.05 |
| | | Wuyungeng 27 | 427.50 | 97.60 | 85.13 | 27.05 | 8.73 | 9.28 |
| | | Mean | 419.48 ± 23.78 a | 91.10 ± 4.18 a | 89.65 ± 4.64 b | 26.73 ± 0.90 b | 8.73 ± 0.44 a | 8.93 ± 0.26 a |
| | LAC | Songzaoxiang 1 | 410.40 | 82.67 | 94.95 | 25.30 | 7.95 | 7.93 |
| | | Yanggeng 239 | 347.70 | 86.70 | 94.78 | 30.65 | 8.42 | 8.14 |
| | | Zaoxianggeng 1 | 370.50 | 80.43 | 95.42 | 28.30 | 7.74 | 7.87 |
| | | Ning 4725 | 401.10 | 88.43 | 95.68 | 25.58 | 8.10 | 7.96 |
| | | Mean | 382.43 ± 24.90 a | 84.56 ± 3.17 a | 95.21 ± 0.36 ab | 27.46 ± 2.18 b | 8.06 ± 0.25 a | 7.98 ± 0.10 a |
| WN | NAC | Jingeng 818 | 288.60 | 69.60 | 97.31 | 31.40 | 6.12 | 5.84 |
| | | Sugeng 815 | 342.00 | 71.50 | 98.21 | 29.93 | 6.45 | 6.64 |
| | | Liangeng 7 | 316.65 | 76.84 | 97.68 | 26.84 | 6.15 | 6.53 |
| | | Wuyungeng 27 | 320.70 | 71.00 | 98.67 | 30.32 | 6.43 | 6.60 |
| | | Mean | 316.99 ± 19.01 b | 72.24 ± 2.75 b | 97.97 ± 0.52 a | 29.62 ± 1.70 a | 6.29 ± 0.15 b | 6.40 ± 0.33 b |
| | LAC | Songzaoxiang 1 | 307.80 | 68.00 | 97.12 | 28.10 | 5.39 | 4.81 |
| | | Yanggeng 239 | 293.70 | 76.30 | 98.05 | 32.44 | 6.23 | 6.17 |
| | | Zaoxianggeng 1 | 324.90 | 75.50 | 94.65 | 28.89 | 6.52 | 6.23 |
| | | Ning 4725 | 404.70 | 71.60 | 91.86 | 26.72 | 6.83 | 6.49 |
| | | Mean | 332.78 ± 42.97 b | 72.85 ± 3.32 b | 95.42 ± 2.40 ab | 29.04 ± 2.11 a | 6.24 ± 0.53 b | 5.93 ± 0.66 b |

[1] NF, nitrogen fertilizer; WN, without nitrogen fertilizer. [2] NAC, *japonica* rice with a normal amylose content; LAC, *japonica* rice with a low amylose content. Different lowercase letters within the same column indicate significantly different means at the 0.05 probability level. The data presented are the mean ± standard deviation, n = 3.

**Table 3.** Milling and appearance quality of *japonica* rice cultivars with different contents of amylose, grown with or without nitrogen fertilizer.

| Nitrogen treatment [1] | Cultivar Type [2] | Cultivar | Milling Quality | | | Appearance Quality | |
|---|---|---|---|---|---|---|---|
| | | | Brown Rice (%) | Milled Rice (%) | Head-Milled Rice (%) | Chalkiness (%) | Chalky Grain Percentage (%) |
| NF | NAC | Jingeng 818 | 85.40 | 76.55 | 72.05 | 4.83 | 18.49 |
| | | Sugeng 815 | 85.65 | 73.80 | 68.03 | 7.57 | 25.10 |
| | | Liangeng 7 | 85.90 | 73.91 | 69.85 | 11.15 | 36.87 |
| | | Wuyungeng 27 | 85.45 | 74.55 | 69.13 | 8.56 | 27.37 |
| | | Mean | 85.60 ± 0.20 a | 74.70 ± 1.10 a | 69.76 ± 1.47 a | 8.03 ± 2.26 a | 26.96 ± 6.59 a |
| | LAC | Songzaoxiang 1 | 83.61 | 76.10 | 71.15 | 12.03 | 51.34 |
| | | Yanggeng 239 | 86.60 | 77.50 | 72.07 | 11.39 | 51.76 |
| | | Zaoxianggeng 1 | 85.25 | 76.35 | 71.37 | 10.63 | 36.79 |
| | | Ning 4725 | 83.55 | 74.50 | 70.55 | 8.93 | 26.85 |
| | | Mean | 84.75 ± 1.27 a | 76.11 ± 1.07 a | 71.28 ± 0.54 a | 10.75 ± 1.16 a | 41.68 ± 10.48 a |
| WN | NAC | Jingeng 818 | 84.50 | 75.75 | 69.05 | 4.37 | 17.39 |
| | | Sugeng 815 | 85.90 | 72.35 | 64.52 | 7.49 | 22.85 |
| | | Liangeng 7 | 85.55 | 73.10 | 66.17 | 7.94 | 29.68 |
| | | Wuyungeng 27 | 85.20 | 74.65 | 70.53 | 8.24 | 24.53 |
| | | Mean | 85.29 ± 0.52 a | 73.96 ± 1.32 a | 67.57 ± 2.36 a | 7.01 ± 1.55 a | 23.61 ± 4.39 a |
| | LAC | Songzaoxiang 1 | 81.95 | 73.25 | 64.82 | 7.63 | 23.96 |
| | | Yanggeng 239 | 84.95 | 75.95 | 69.75 | 13.35 | 47.80 |
| | | Zaoxianggeng 1 | 84.35 | 74.70 | 71.79 | 8.55 | 35.28 |
| | | Ning 4725 | 82.05 | 72.70 | 69.52 | 7.38 | 29.19 |
| | | Mean | 83.33 ± 1.34 a | 74.15 ± 1.27 a | 68.97 ± 2.55 a | 9.23 ± 2.42 a | 34.05 ± 8.89 a |

[1] NF, nitrogen fertilizer; WN, without nitrogen fertilizer. [2] NAC, *japonica* rice with a normal amylose content; LAC, *japonica* rice with a low amylose content. Different lowercase letters within the same column indicate significantly different means at the 0.05 probability level. The data presented are the mean ± standard deviation, n = 3.

### 3.3. Eating Qualities and Their Relationships

Compared with that in WN, the amylose content was significantly lower in both types of rice cultivars in NF (Table 4). By contrast, the protein content was higher in NF than in WN. In both WN and NF, the protein content was higher in NAC than in LAC cultivars. In a comparison of the variability in the amylose content between LAC and NAC cultivars in NF, the coefficient of variation (CV) of the NAC cultivars (CV = 5.09%) was higher than that of the LAC cultivars (CV = 4.49%). Therefore, the response of the amylose content of the NAC cultivars to N fertilization was more variable.

**Table 4.** Protein content, amylose content, and eating quality of *japonica* rice cultivars with different contents of amylose, grown with or without nitrogen fertilizer.

| Nitrogen Treatment [1] | Cultivar Type [2] | Cultivar | Protein Content (%) | Amylose Content (%) | Hardness | Stickiness | Palatability |
|---|---|---|---|---|---|---|---|
| NF | NAC | Jingeng 818 | 9.50 | 16.30 | 8.80 | 2.65 | 42.00 |
| | | Sugeng 815 | 9.80 | 15.45 | 8.10 | 3.45 | 48.75 |
| | | Liangeng 7 | 9.45 | 16.55 | 8.53 | 2.73 | 43.75 |
| | | Wuyungeng 27 | 9.05 | 17.80 | 8.15 | 3.65 | 49.50 |
| | | Mean | 9.45 ± 0.27 a | 16.53 ± 0.84 b | 8.39 ± 0.29 a | 3.12 ± 0.44 b | 46.00 ± 3.20 c |
| | LAC | Songzaoxiang 1 | 9.05 | 13.15 | 7.30 | 5.30 | 60.50 |
| | | Yanggeng 239 | 9.00 | 12.85 | 7.13 | 5.18 | 61.50 |
| | | Zaoxianggeng 1 | 8.30 | 14.25 | 6.58 | 7.08 | 66.50 |
| | | Ning 4725 | 8.10 | 14.15 | 6.23 | 6.78 | 64.24 |
| | | Mean | 8.61 ± 0.42 b | 13.60 ± 0.61 c | 6.81 ± 0.43 b | 6.08 ± 0.85 a | 63.19 ± 5.52 ab |
| WN | NAC | Jingeng 818 | 8.60 | 18.80 | 8.35 | 2.68 | 44.50 |
| | | Sugeng 815 | 8.45 | 18.15 | 7.65 | 4.16 | 53.63 |
| | | Liangeng 7 | 8.85 | 17.65 | 7.95 | 3.86 | 51.13 |
| | | Wuyungeng 27 | 8.15 | 20.05 | 7.71 | 4.28 | 53.88 |
| | | Mean | 8.51 ± 0.25 b | 18.66 ± 0.90 a | 7.92 ± 0.27 a | 3.74 ± 0.64 b | 50.78 ± 3.78 bc |
| | LAC | Songzaoxiang 1 | 8.65 | 15.95 | 6.28 | 7.34 | 73.38 |
| | | Yanggeng 239 | 8.00 | 16.15 | 6.30 | 6.26 | 70.00 |
| | | Zaoxianggeng 1 | 8.10 | 16.25 | 6.21 | 6.89 | 72.50 |
| | | Ning 4725 | 7.80 | 16.55 | 6.10 | 7.33 | 74.50 |
| | | Mean | 8.14 ± 0.31 b | 16.23 ± 0.22 b | 6.22 ± 0.08 b | 6.95 ± 0.44 a | 72.59 ± 1.66 a |

[1] NF, nitrogen fertilizer; WN, without nitrogen fertilizer. [2] NAC, *japonica* rice with a normal amylose content; LAC, *japonica* rice with a low amylose content. Different lowercase letters within the same column indicate significantly different means at the 0.05 probability level. The data presented are the mean ± standard deviation, n = 3.

In WN and NF, the LAC cultivars had the highest palatability, indicating the best eating quality (Table 4). Compared with that in WN, the palatability was lower in both types of rice cultivars in NF. Regardless of the N level, the LAC cultivars had significantly higher palatability than the NAC cultivars. Higher palatability was associated with lower hardness and higher stickiness. Compared with the NAC cultivars, the LAC cultivars had lower hardness and higher stickiness.

The amylose content was negatively correlated with the palatability and protein content (Table 5). The protein content was significantly negatively correlated with palatability and stickiness, but significantly positively correlated with hardness. Hence, the low amylose content and low protein content both contributed to the higher palatability of the cooked rice.

### 3.4. Pasting Properties in Relation to the Eating Quality

The differences in the pasting properties between the two types of *japonica* rice cultivars in the two N treatments are shown in Table 6. In NF, the peak viscosity, trough viscosity, breakdown and final viscosity decreased compared with those in WN. Regardless of the N treatment, compared with the NAC cultivars, the peak viscosity and breakdown increased, whereas the final viscosity, setback and pasting temperature decreased in the LAC cultivars.

The amylose content was significantly negatively correlated with breakdown, and was significantly positively correlated with final viscosity (Table 5). Palatability was significantly positively correlated with the peak viscosity and breakdown, and was significantly negatively correlated with the final viscosity and setback. Therefore, the characteristics

of high eating quality were as follows: a low amylose content, a high peak viscosity and breakdown, and a low final viscosity and setback.

**Table 5.** Correlation matrix of the protein content, amylose content, eating quality and pasting properties [1] of *japonica* rice cultivars.

|     | AC      | PA         | HN         | SN         | PV         | TV       | BD         | FV         | SB         | PT      |
|-----|---------|------------|------------|------------|------------|----------|------------|------------|------------|---------|
| PC  | −0.164  | −0.726 **  | 0.756 **   | −0.704 **  | −0.587 *   | −0.480   | −0.320     | −0.019     | 0.376      | 0.231   |
| AC  |         | −0.494 *   | 0.400      | −0.439     | −0.083     | 0.368    | −0.513 *   | 0.775 **   | 0.693 **   | 0.141   |
| PA  |         |            | −0.994 **  | 0.996 **   | 0.596 *    | 0.116    | 0.723 **   | −0.543 *   | −0.845 **  | −0.254  |
| HN  |         |            |            | −0.982 **  | −0.584 *   | −0.123   | −0.700 **  | 0.532 *    | 0.829 **   | 0.222   |
| SN  |         |            |            |            | 0.615 *    | 0.118    | 0.749 **   | −0.535 *   | −0.851 **  | −0.260  |
| PV  |         |            |            |            |            | 0.713 ** | 0.658 **   | 0.113      | −0.575 *   | −0.215  |
| TV  |         |            |            |            |            |          | −0.059     | 0.709 **   | 0.108      | −0.426  |
| BD  |         |            |            |            |            |          |            | −0.601*    | −0.934 **  | 0.152   |
| FV  |         |            |            |            |            |          |            |            | 0.748 **   | −0.210  |
| SB  |         |            |            |            |            |          |            |            |            | −0.029  |

[1] PC, protein content; AC, amylose content; PA, palatability; HN, hardness; SN, stickiness; PV, peak viscosity; TV, trough viscosity; BD, breakdown; FV, final viscosity; SB, setback; PT, pasting temperature. * and ** indicate significance at the $p < 0.05$ and $p < 0.01$ levels, respectively (n = 16).

**Table 6.** Pasting properties of *japonica* rice cultivars with different contents of amylose, grown with or without nitrogen fertilizer.

| Nitrogen Treatment [1] | Cultivar Type [2] | Cultivar | Peak Viscosity (cP) | Trough Viscosity (cP) | Breakdown (cP) | Final Viscosity (cP) | Setback (cP) | Pasting Temperature (°C) |
|---|---|---|---|---|---|---|---|---|
| NF | NAC | Jingeng 818 | 2401.0 | 1307.0 | 1094.0 | 2465.0 | 64.0 | 70.45 |
|    |     | Sugeng 815 | 2244.0 | 970.0 | 1274.0 | 1844.5 | −399.5 | 75.58 |
|    |     | Liangeng 7 | 2200.5 | 1124.0 | 1076.5 | 2262.0 | 61.5 | 69.33 |
|    |     | Wuyungeng 27 | 2427.0 | 1374.0 | 1053.0 | 2546.0 | 119.0 | 69.70 |
|    |     | Mean | 2318.1 ± 97.5 b | 1193.8 ± 158.3 b | 1124.4 ± 87.6 a | 2279.4 ± 271.6 ab | −38.8 ± 209.5 a | 71.26 ± 2.5 a |
|    | LAC | Songzaoxiang 1 | 3107.0 | 1516.0 | 1591.0 | 2124.0 | −983.0 | 69.75 |
|    |     | Yanggeng 239 | 2570.5 | 1321.0 | 1249.5 | 1947.5 | −623.0 | 70.10 |
|    |     | Zaoxianggeng 1 | 2928.0 | 1440.5 | 1487.5 | 2097.5 | −830.5 | 70.50 |
|    |     | Ning 4725 | 2692.5 | 1307.5 | 1385.0 | 1926.5 | −766.0 | 70.10 |
|    |     | Mean | 2824.5 ± 207.6 a | 1396.3 ± 86.4 ab | 1428.3 ± 126.3 a | 2023.9 ± 87.7 b | −800.6 ± 129.3 b | 70.11 ± 0.3 a |
| WN | NAC | Jingeng 818 | 2952.0 | 1904.0 | 1048.0 | 3078.0 | 126.0 | 71.35 |
|    |     | Sugeng 815 | 2717.0 | 1219.5 | 1497.5 | 2188.5 | −528.5 | 75.58 |
|    |     | Liangeng 7 | 2723.5 | 1556.5 | 1167.0 | 2753.0 | 29.5 | 69.73 |
|    |     | Wuyungeng 27 | 2742.0 | 1676.0 | 1066.0 | 2890.0 | 148.0 | 70.08 |
|    |     | Mean | 2783.6 ± 97.6 a | 1589.0 ± 247.2 a | 1194.6 ± 180.7 a | 2727.4 ± 331.8 a | −56.3 ± 276.3 a | 71.68 ± 2.3 a |
|    | LAC | Songzaoxiang 1 | 3242.0 | 1565.5 | 1676.5 | 2221.5 | −1020.5 | 70.08 |
|    |     | Yanggeng 239 | 2765.5 | 1473.5 | 1292.0 | 2169.5 | −596.0 | 70.13 |
|    |     | Zaoxianggeng 1 | 2709.0 | 1367.0 | 1342.0 | 2018.0 | −691.0 | 69.65 |
|    |     | Ning 4725 | 2931.0 | 1473.0 | 1458.0 | 2125.0 | −806.0 | 70.40 |
|    |     | Mean | 2911.9 ± 207.3 a | 1469.8 ± 70.3 ab | 1442.1 ± 148.1 a | 2133.5 ± 74.9 b | −778.4 ± 158.3 b | 70.06 ± 0.3 a |

[1] NF, nitrogen fertilizer; WN, without nitrogen fertilizer. [2] NAC, *japonica* rice with a normal amylose content; LAC, *japonica* rice with a low amylose content. Different lowercase letters within the same column indicate significantly different means at the 0.05 probability level. The data presented are the mean ± standard deviation, n = 3.

### 3.5. Thermal Properties of Starch

The starch of both types of rice cultivars in both N treatments showed no significant differences in its gelatinization temperatures and enthalpy (ΔHgel) (Table 7). Compared with those in WN, in NF the retrogradation enthalpy (ΔHret) and degree (%R) increased by 51.6% and 48.8% in the NAC cultivars, and by 31.9% and 30.0% in the LAC cultivars, respectively. Therefore, NF increased the starch ΔHret and %R. Compared with those in the LAC cultivars, ΔHret and %R in the NAC cultivars increased by 49.2% and 55.2% in NF, and by 29.8% and 35.6% in WN, respectively. Hence, the ΔHret and %R were higher in NAC than in the LAC cultivars.

The amylose content was significantly negatively correlated with ΔHgel (Table 8). Palatability and stickiness were both significantly negatively correlated with ΔHret and %R. Hardness was significantly positively correlated with ΔHret and %R.

**Table 7.** Thermal properties of *japonica* rice cultivar starches with different contents of amylose, grown with or without nitrogen fertilization.

| Nitrogen Treatment [1] | Cultivar Type [2] | Cultivar | $T_o$ (°C) | $T_p$ (°C) | $T_c$ (°C) | $\Delta H$gel (J/g) | $\Delta H$ret (J/g) | %R |
|---|---|---|---|---|---|---|---|---|
| NF | NAC | Jingeng 818 | 60.55 | 65.40 | 73.10 | 11.02 | 2.27 | 20.59 |
| | | Sugeng 815 | 66.15 | 71.75 | 78.40 | 12.43 | 2.33 | 18.75 |
| | | Liangeng 7 | 61.50 | 66.90 | 74.80 | 11.50 | 1.69 | 14.66 |
| | | Wuyungeng 27 | 61.90 | 66.45 | 73.50 | 10.92 | 1.13 | 10.31 |
| | | Mean | 62.53 ± 2.15 a | 67.63 ± 2.44 a | 74.95 ± 2.09 a | 11.47 ± 0.60 a | 1.85 ± 0.49 a | 16.16 ± 3.96 a |
| | LAC | Songzaoxiang 1 | 63.95 | 68.15 | 74.10 | 12.49 | 1.46 | 11.69 |
| | | Yanggeng 239 | 63.45 | 68.20 | 74.80 | 11.72 | 1.17 | 10.00 |
| | | Zaoxianggeng 1 | 63.10 | 67.90 | 75.00 | 11.69 | 1.30 | 11.16 |
| | | Ning 4725 | 63.55 | 68.35 | 75.85 | 11.61 | 1.01 | 8.69 |
| | | Mean | 63.51 ± 0.30 a | 68.15 ± 0.16 a | 74.94 ± 0.62 a | 11.88 ± 0.36 a | 1.24 ± 0.17 ab | 10.41 ± 1.15 ab |
| WN | NAC | Jingeng 818 | 61.60 | 67.15 | 74.75 | 11.40 | 1.09 | 9.54 |
| | | Sugeng 815 | 67.25 | 72.30 | 75.85 | 11.87 | 1.46 | 12.27 |
| | | Liangeng 7 | 61.65 | 66.30 | 73.80 | 10.77 | 1.17 | 10.88 |
| | | Wuyungeng 27 | 61.55 | 66.50 | 74.15 | 10.82 | 1.16 | 10.70 |
| | | Mean | 63.01 ± 2.45 a | 68.06 ± 2.47 a | 74.64 ± 0.78 a | 11.21 ± 0.45 a | 1.22 ± 0.14 ab | 10.86 ± 0.97 ab |
| | LAC | Songzaoxiang 1 | 63.55 | 67.80 | 73.80 | 12.04 | 0.72 | 5.96 |
| | | Yanggeng 239 | 63.05 | 67.85 | 75.40 | 11.63 | 0.70 | 6.02 |
| | | Zaoxianggeng 1 | 62.75 | 67.45 | 74.95 | 11.15 | 1.29 | 11.61 |
| | | Ning 4725 | 63.55 | 68.30 | 75.55 | 11.95 | 1.03 | 8.64 |
| | | Mean | 63.23 ± 0.34 a | 67.85 ± 0.30 a | 74.93 ± 0.69 a | 11.69 ± 0.35 a | 0.94 ± 0.25 b | 8.01 ± 2.32 b |

[1] NF, nitrogen fertilizer; WN, without nitrogen fertilizer. [2] NAC, *japonica* rice with a normal amylose content; LAC, *japonica* rice with a low amylose content. To, Tp, Tc, $\Delta H$gel, $\Delta H$ret and %R correspond to onset temperature, peak temperature, conclusion temperature, gelatinization enthalpy, and retrogradation enthalpy and degree, respectively. Different lowercase letters within the same column indicate significantly different means at the 0.05 probability level. The data presented are the mean ± standard deviation, n = 3.

**Table 8.** Correlations between the eating quality, starch structure and gelatinization properties [1] of *japonica* rice cultivars.

| | $T_O$ | $T_P$ | $T_C$ | $\Delta H$gel | $\Delta H$ret | %R | PI | PWHM | PP | LD | RC |
|---|---|---|---|---|---|---|---|---|---|---|---|
| AC | −0.261 | −0.175 | −0.186 | −0.587 * | −0.084 | −0.011 | −0.546 * | −0.041 | −0.499 * | 0.32 | −0.740 ** |
| PA | 0.267 | 0.137 | 0.143 | 0.318 | −0.614 * | −0.656 ** | 0.709 ** | 0.494 | 0.216 | −0.121 | 0.208 |
| HN | −0.292 | −0.171 | −0.188 | −0.316 | 0.640 ** | 0.683 ** | −0.725 ** | −0.486 | −0.24 | 0.131 | −0.188 |
| SN | 0.256 | 0.12 | 0.107 | 0.32 | −0.590 * | −0.630 ** | 0.681 ** | 0.496 | 0.174 | −0.11 | 0.203 |

[1] AC, amylose content; PA, palatability; HN, hardness; SN, stickiness; $T_O$, onset temperature; $T_p$, peak temperature; $T_c$, conclusion temperature; $\Delta H$gel, gelatinization enthalpy; $\Delta H$ret, retrogradation enthalpy; %R, retrogradation degree; PI, peak intensity; PWHM, peak width at half maximum; PP, peak position; LD, lamellar distance; RC, relative crystallinity. * and ** indicate significance at the $p < 0.05$ and $p < 0.01$ levels, respectively (n = 16).

### 3.6. Small-Angle X-ray Scattering Variables and the Relative Crystallinity of the Starches

The XRD patterns of the starch of eight rice cultivars affected by nitrogen levels were very similar (Figure 2), showing the typical A-type diffraction pattern. The relative crystallinity calculated from the XRD patterns showed no significant differences between the nitrogen levels or the type of rice cultivar (Table 9). Compared with that in WN, the relative crystallinity increased by 5.0% in the NAC cultivars, and by 3.6% in the LAC cultivars in NF. The relative crystallinity was 2.8% and 4.1% lower in NAC cultivars than in LAC cultivars in NF and WN, respectively. The amylose content was significantly negatively correlated with the relative crystallinity (Table 8). Therefore, the starch samples with a low amylose content had a high relative crystallinity.

The lamellar structures of rice starch were investigated using SAXS (Figure 3). The peak intensity, peak width at half maximum, peak position, and lamellar distance calculated from the SAXS patterns are shown in Table 9. The SAXS variables of both cultivar types were not significantly different between NF and WN. Compared with that in the LAC cultivars, the peak intensity in the NAC cultivars decreased by 17.9% in NF and by 21.6% in WN. The peak width at half maximum, peak position and lamellar distance were similar between the NAC and LAC cultivars in the two N treatments. The peak intensity and peak position were significantly negatively correlated with the amylose content, and the peak intensity was significantly positively correlated with the palatability and stickiness (Table 8).

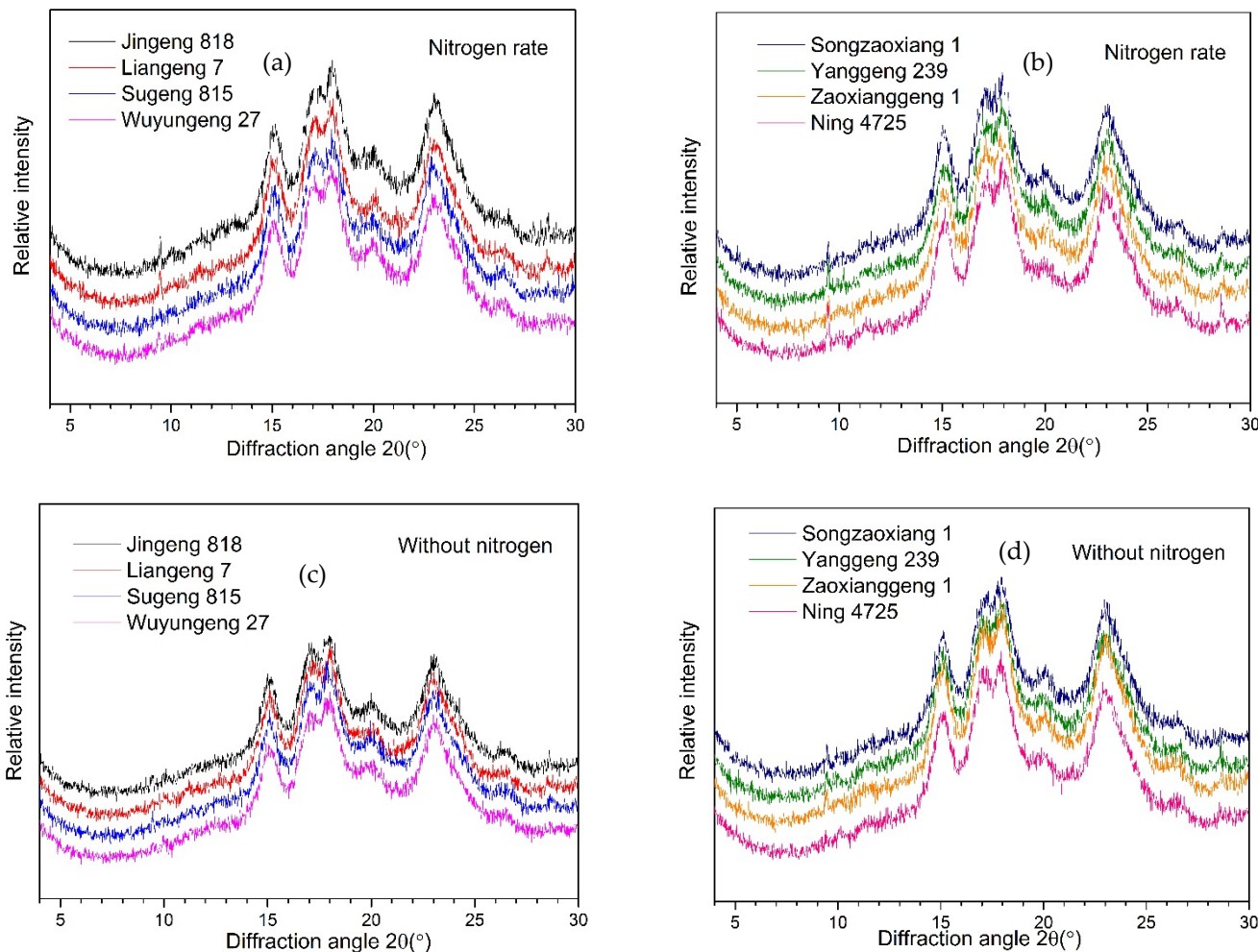

**Figure 2.** X-ray diffraction patterns of the starch of *japonica* rice cultivars with different contents of amylose, with or without nitrogen fertilization. (**a**) NAC, with fertilization; (**b**) LAC, with fertilization; (**c**) NAC, without fertilization; (**d**) LAC, without fertilization.

**Table 9.** Small-angle X-ray scattering (SAXS) variables and relative crystallinity of the starch of *japonica* rice cultivars with different contents of amylose, grown with or without nitrogen fertilizer.

| Nitrogen Treatment [1] | Cultivar Type [2] | Cultivar | SAXS Variable | | | | Relative Crystallinity (%) |
|---|---|---|---|---|---|---|---|
| | | | Peak Intensity (Count) | Peak Width at Half Maximum (Å$^{-1}$) | Peak Position (Å$^{-1}$) | Lamellar Distance (nm) | |
| NF | NAC | Jingeng 818 | 108.28 | 0.019 | 0.067 | 9.38 | 30.4 |
| | | Sugeng 815 | 120.56 | 0.017 | 0.069 | 9.13 | 32.3 |
| | | Liangeng 7 | 130.55 | 0.019 | 0.068 | 9.24 | 33.9 |
| | | Wuyungeng 27 | 124.30 | 0.019 | 0.068 | 9.24 | 30.4 |
| | | Mean | 120.92 ± 8.12 b | 0.018 ± 0.00 a | 0.068 ± 0.00 a | 9.25 ± 0.09 a | 31.75 ± 1.46 a |
| | | Songzaoxiang 1 | 150.34 | 0.018 | 0.068 | 9.19 | 33 |
| | | Yanggeng 239 | 157.66 | 0.020 | 0.070 | 9.01 | 33.3 |
| | | Zaoxianggeng 1 | 128.87 | 0.019 | 0.068 | 9.19 | 32.9 |
| | | Ning 4725 | 152.40 | 0.018 | 0.069 | 9.38 | 31.5 |
| | | Mean | 147.32 ± 10.98 a | 0.019 ± 0.00 a | 0.069 ± 0.00 a | 9.19 ± 0.13 a | 32.68 ± 0.69 a |

**Table 9.** *Cont.*

| Nitrogen Treatment [1] | Cultivar Type [2] | Cultivar | SAXS Variable | | | | Relative Crystallinity (%) |
|---|---|---|---|---|---|---|---|
| | | | Peak Intensity (Count) | Peak Width at Half Maximum ($\mathrm{\mathring{A}}^{-1}$) | Peak Position ($\mathrm{\mathring{A}}^{-1}$) | Lamellar Distance (nm) | |
| WN | NAC | Jingeng 818 | 123.24 | 0.018 | 0.068 | 9.23 | 29.7 |
| | | Sugeng 815 | 129.48 | 0.018 | 0.068 | 9.19 | 31.6 |
| | | Liangeng 7 | 98.61 | 0.019 | 0.068 | 9.21 | 30.5 |
| | | Wuyungeng 27 | 118.06 | 0.019 | 0.068 | 9.28 | 29.2 |
| | | Mean | 117.35 ± 11.55 b | 0.018 ± 0.00 a | 0.068 ± 0.00 a | 9.23 ± 0.03 a | 30.25 ± 0.91 a |
| | LAC | Songzaoxiang 1 | 146.48 | 0.020 | 0.068 | 9.21 | 32.1 |
| | | Yanggeng 239 | 153.79 | 0.020 | 0.068 | 9.21 | 32.5 |
| | | Zaoxianggeng 1 | 132.39 | 0.020 | 0.068 | 9.23 | 31.0 |
| | | Ning 4725 | 166.19 | 0.020 | 0.068 | 9.19 | 30.5 |
| | | Mean | 149.71 ± 12.23 a | 0.020 ± 0.00 a | 0.068 ± 0.00 a | 9.21 ± 0.01 a | 31.53 ± 0.81 a |

[1] NF, nitrogen fertilizer; WN, without nitrogen fertilizer. [2] NAC, *japonica* rice with a normal amylose content; LAC, *japonica* rice with a low amylose content. Different lowercase letters within the same column indicate significantly different means at the 0.05 probability level. The data presented are the mean ± standard deviation, n = 3.

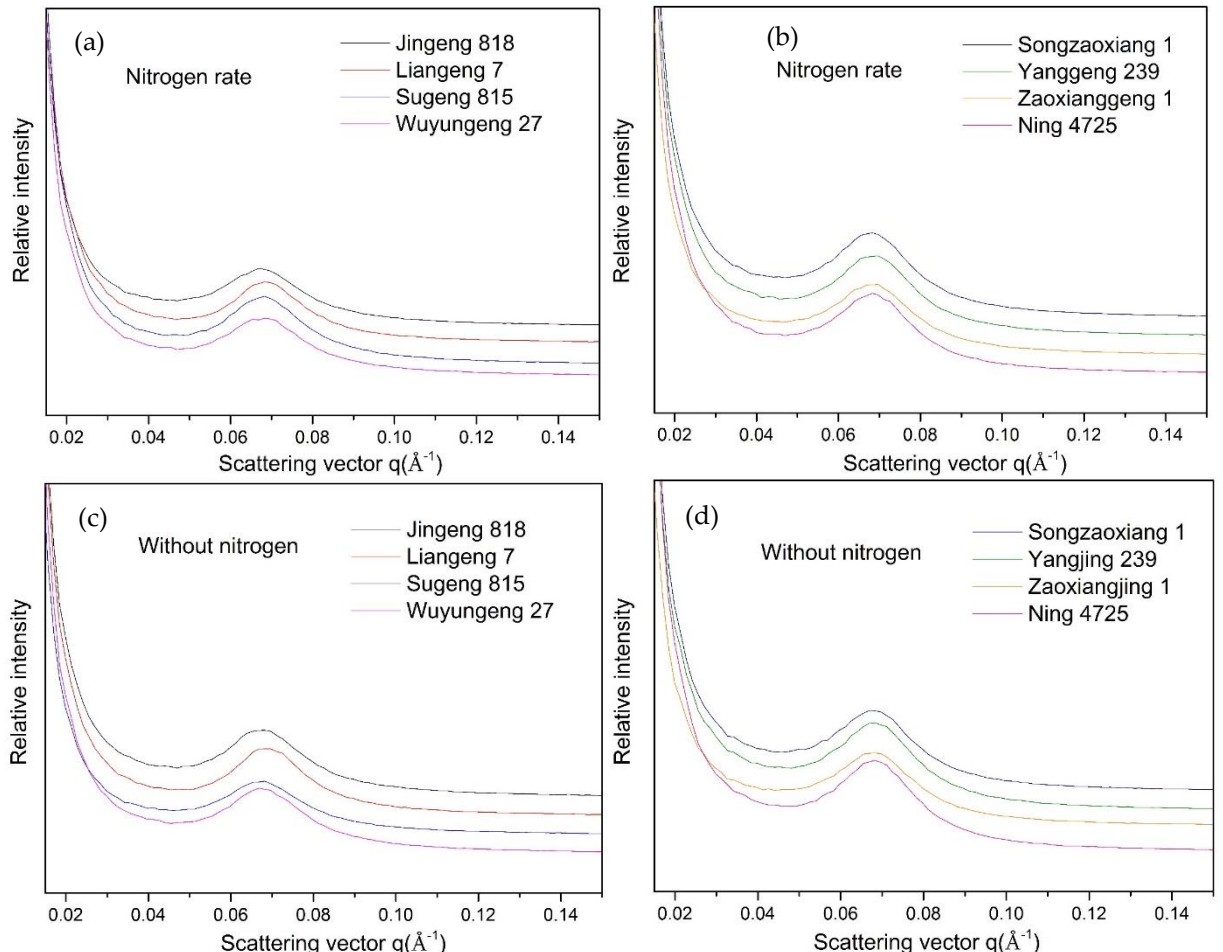

**Figure 3.** Small-angle X-ray scattering spectra of the starch of *japonica* rice cultivars with different contents of amylose, with or without nitrogen fertilization. (**a**) NAC, with fertilization; (**b**) LAC, with fertilization; (**c**) NAC, without fertilization; (**d**) LAC, without fertilization.

## 4. Discussion

Rice grain quality is markedly influenced by the cultivar genotype and N application [24]. In previous studies, N fertilizer application improved the milling quality and nutritional quality of rice [11,25]. Our findings are consistent with these results, and in

NF, the milling quality and nutritional quality improved, although the appearance quality and eating quality deteriorated compared to WN (Tables 2 and 3). Top-dressed nitrogen increases the protein content but decreases the amylose content in rice grains [26,27]. Similarly, in our study, the protein content was higher and the amylose content was lower in NF than in WN. Compared with WN, the palatability of the cooked rice decreased significantly in NF because of the higher protein content and hardness and the lower stickiness. Ref. [28] reported similar results in which a high protein content led to a firmer texture of rice, resulting in the insufficient absorption of water for compete gelatinization. Differences in grain quality are found in rice cultivars with different contents of amylose [9]. Zhao et al. [18] reported that rice varieties with lower amylose contents had a lower peak time, trough, final, setback, and consistent viscosity, as well as a lower hardness of the cooked rice and a higher gel consistency, breakdown viscosity, stickiness and comprehensive value of the cooked rice. In this study, compared with those of NAC cultivars, the peak viscosity, breakdown value and stickiness were higher, and the final viscosity, setback value, protein content and hardness were lower in LAC cultivars, leading to the higher palatability of cooked rice.

Yang et al. [29] examined the response of starch physicochemical properties to N application, and according to Zhu et al. [30], an increasing level of N decreased the gelatinization temperature and increased the relative crystallinity and gelatinization enthalpy. Those results are partially consistent with our findings. The highest relative crystallinity (Table 9), gelatinization enthalpy and retrogradation enthalpy (Table 7) were observed in NF, compared with WN. Singh et al. [31] also found that the amylose content decreased, and the pasting temperature, gelatinization temperature and enthalpy increased with N application.

The starch physicochemical properties of different types of rice cultivar have been studied extensively. Cai et al. [15] investigated the structure and functional properties of the starch of 10 rice cultivars with different amylose contents, and found that an increased amylose content increased the gelatinization enthalpy and decreased the relative crystallinity and peak intensity. Compared with NAC cultivars, the relative crystallinity and peak intensity increased (Table 9), and the retrogradation enthalpy and degree decreased (Table 7) in LAC cultivars. However, the gelatinization temperature and enthalpy were similar between LAC and NAC cultivars (Table 7). The intensity of the scattering peaks depends primarily on the order degree in semicrystalline regions, and decreases with an increasing amylose content [32]. Thus, our findings are consistent with those of previous studies [15,32].

The amylose content and protein content in the rice endosperm are the main factors that affect the rice eating quality [28,33,34]. In this study, the amylose content and protein content were significantly negatively correlated with the palatability of cooked rice. According to previous studies, the difference in the amylose content determines the starch crystalline structure and physicochemical properties [35–37]. Kong et al. [35] found that the amylose content was significantly positively correlated with the peak viscosity, hot paste viscosity, cold paste viscosity, setback and hardness, whereas it was negatively correlated with adhesiveness, cohesiveness, gelatinization temperature (*To*, *Tp*, *Tc*) and enthalpy. In our study, the amylose content was significantly positively correlated with the final viscosity and setback, and was significantly negatively correlated with the relative crystallinity, peak intensity, gelatinization enthalpy and breakdown (Table 8); these results are consistent with those of previous studies [15,35].

Pasting properties are also closely associated with the rice eating quality. Our findings showed significant positive correlations between palatability, peak viscosity and breakdown, and significant negative correlations between palatability, final viscosity and setback (Table 5). These results support those of previous studies in which rice with a high eating quality had a high peak viscosity and breakdown, and a low setback (Asante et al., 2013; Zhao et al., 2019). Rice cultivars with low gelatinization temperatures and enthalpy require less water and cooking time, and absorb less thermal energy to reach starch gelatiniza-

tion, contributing to the high palatability of the cooked rice [38]. In our study, there was no significant correlation between the palatability and gelatinization temperature or enthalpy, but palatability was significantly negatively correlated with the retrogradation enthalpy and degree (Table 8). These results are partially consistent with those of a previous study [33].

Therefore, in this study, the high eating quality of rice was associated with a higher peak viscosity, breakdown, relative crystallinity and peak intensity, as well as a lower protein content, amylose content, final viscosity, setback, and retrogradation enthalpy and degree.

## 5. Conclusions

Compared with no N application, N fertilizer led to the deterioration of the eating quality of rice cultivars because of the higher protein content, lower pasting viscosity, and higher retrogradation enthalpy and degree. Compared with NAC cultivars, LAC cultivars had a higher pasting viscosity and lower retrogradation enthalpy and degree, leading to a higher eating quality; however, they also tended to have an inferior appearance quality. The amylose content and protein content were significantly negatively correlated with the palatability of cooked rice. The differences in the amylose content determined the starch crystal structure and pasting and gelatinization properties. Significant correlations were detected between the amylose content, eating quality and starch properties. Therefore, in order to increase the eating quality of cooked rice, an LAC rice cultivar should be selected and grown under N deficiency because the amylose content and protein content decrease, and the pasting properties and retrogradation properties improve. These results provide new insight into the grain quality of rice and how to direct breeding and agronomic management to achieve high eating quality.

**Author Contributions:** Conceptualization, Y.H. and H.Z.; methodology, Y.H.; software, Y.H.; validation, Y.H., S.C. and H.Z.; formal analysis, Y.H.; investigation, Y.H.; resources, H.Z.; data curation, S.C.; writing—original draft preparation, Y.H. and S.C.; writing—review and editing, Y.H. and H.Z.; visualization, Y.H. and H.Z.; supervision, H.Z.; project administration, Y.H. and H.Z.; funding acquisition, Y.H. and H.Z. All authors have read and agreed to the published version of the manuscript.

**Funding:** This research was funded by National Nature Science Foundation of China (31701350), National Key Research Program of China (2016YFD0300503, 2017YFD0301205), the Earmarked Fund for China Agriculture Research System (CARS-01-27), and the Earmarked Fund for Jiangsu Agriculture Industry Technology System (JATS [2019]444) and the project by the Priority Academic Program Development of Jiangsu Higher Education Institutions.

**Institutional Review Board Statement:** Not applicable.

**Informed Consent Statement:** Not applicable.

**Data Availability Statement:** Not applicable.

**Conflicts of Interest:** The authors declare no conflict of interest.

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
