# Peer review of "Comparison of the Grain Quality and Starch Physicochemical Properties between Japonica Rice Cultivars with Different Contents of Amylose, as Affected by Nitrogen Fertilization"

_agriculture, doi:10.3390/agriculture11070616_

Round 1

Reviewer 1 Report

Reviewed manuscript entitled „Comparison of grain quality and starch physicochemical properties between japonica rice cultivars with different contents of amylose as affected by nitrogen fertilization” is an original research study.

Following are the detailed suggestions and questions for the Authors:

Materials and methods

What was the selection criterion for the cultivars/lines for the experiment? I request information.

No information about the weather as it was during the rice growing season. Maybe it is worth mentioning in a few sentences what was the average temperature, rainfall in 2016 and 2017.

Line 97-100, which method was used to test soil nutrient contents?

Conclusions

It would be better if you could ended your conclusion with one or two sentences regarding the vision of the future of your studies.

Reference

Reference 10 (line 444) - is not quoted. Please correct or complete.

Reference 26 (line 481) - is not quoted. Please correct or complete.

Author Response

Dear reviewer,

       Thank you very much for reviewing this paper. According the comments and suggestions, we revise the paper as follows.

1. We have two criterion for the selected cultivars/lines. The first selection criterion is amylose content of rice cultivar, and the second selection criterion is the similar growth period.

2. We add the weather information during the rice growing season.

3. The method of testing soil nutrient contents is added in this paper.

4.  We prospect the vision of the future in the conclusion. We think that this study provide new insight into grain quality of rice and how to take breeding and agronomic management to achieve high eating quality.

5.  Sorry, the error of the reference 10 and 26 is not found.

Thank you

Yajie Hu

Reviewer 2 Report

The manuscript is well written and the topic is current and interesting. However, the discussion should be more elaborate than the current. The authors emphasize quality traits more than discussing the influence of N on these traits.

Below are a few comments

Introduction

The statement, “Nitrogen (N) is a crucial factor that affects AC and grain quality and the starch physicochemical properties. In previous studies, the amylose content decreased with an increase in the N level in both NAC (Gao et al. 2010) and LAC (Zhu et al. 2017a) rice cultivars. How the susceptibility of AC to N deficiency is affected by NAC and LAC cultivars is unclear.” Should be introduced in a new paragraph.

In addition, more information (with relevant references) should be included on how different N fertilizations influence rice quality.

Methods

Why was the grains for 2017 considered for quality assessment? Don’t you think weather can have influence on quality of crops? Please provide reason why you used the 2016 data instead of average of all.

The discussion should include information that can answer the following questions

  • What is the implication if N fertilizers did reduced quality? Does it mean farmers should half the fertilizers or zero? Then this should also illustrate clearly how can we account for the tradeoff between yield and quality in relation to the N?
  • In addition, we should clearly illustrate the mechanisms through which amylose content is reduced when N is applied for the cultivation of these new rice cultivars? This should be connected to how excess N influence the eating quality of crops (rice).
  • The influence of genetic factors should not be under looked in this study. Please provide detailed explanations about how genes influence quality in those different rice cultivars. This is because overall performance of crops is as a result of gene and environment interactions. If we controlled environment by supplying varying rates of N fertilizer, then unexplained phenomenon in this study should clearly be attributed to the genetic constitutions of the cultivars. Please consider adding genetic variation among cultivars.

Thank you.

Author Response

Dear reviewer,

       Thank you very much for reviewing this paper. According the comments and suggestions, we revise the paper as follows.

1. The statement about effect of N application on grain quality have been added in a new paragraph.

2.The similar mean temperature during rice growing was shown in two years. There was no significant difference in grain quality between 2016 and 2017. We only use the 2016 data due to lack of the portion of starch measurement data of 2017.

3.We find that N fertilizer application increased the processing and nutritional quality, but decreased the appearance and eating quality in this study. Above all, N fertilizer application increased rice yield. For farmers, the high yield of rice is the most important. Numerous studies have reported less or zero nitrogen application could improve the eating quality of rice. Especially, zero chemistry fertilizer (include N) is used in the production of organic rice. Hence, the farmers select different N rate according to the target of grain yield and quality.

4. When rice increased N fertilizer application, the nitrogen metabolism is enhanced, on the contrary, the carbon metabolism is reduced. Hence, the amylose content reduced when N is applied for rice.

5. In this study, the different type rice cultivar was classified into low amylose content and normal amylose content. The gene that control amylose synthesis is mainly Wx. In this study, we only used two N rate, so the varying rates of N fertilizer should be used in the further study.

Thank you

Yajie Hu

Reviewer 3 Report

The work analyzes the effects of nitrogen fertilization on 8 rice cultivars with contrasting amylose content , taking into consideration different aspects of grain quality  and of starch characteristics. 

In my opinion, the major weakness of the study is that it is not well defined how some quality parameters have been assessed as for ex. sensory properties, milling quality, palatability, appearence quality and nutritional traits.

In detail: 

Line 105: Please delete 'detailed'

Line 182: Who did determine the sensory properties? Please specify the methodology

Line 208: What do the Authors intend for 'milling quality'? is it the flour yield? Please specify

Line 277-236: How palatability parameter has been determined?

Lines 256-260, Table 8, 374-375, 386,395 : same comment as above

Line 336: Which are the nutritional parameters that the Authors investigated?

Line 406: The appearance quality shoul be more exstensively defined

Author Response

Dear reviewer,

       Thank you very much for reviewing this paper. According the comments and suggestions, we revise the paper as follows.

1. We defined grain quality parameters in materials and method, such as processing quality and eating quality.

2. We delete 'detailed' in Line 105

3. In this study, we determine the sensory properties (palatability, hardness, stickiness) by using an STA-1A rice sensory analyzer. The methodology is specified.

4. In this study, the milling quality is not flour yield, we have change the statement “milling quality” to “processing quality”.

5. The palatability parameter is determined by an STA-1A rice sensory analyzer, which reflecting the eating quality.

6. In this study, we investigated the protein content, which indicates the nutritional parameters .

Thank you.

Yajie Hu

Round 2

Reviewer 2 Report

Dear Authors,

Thank you for addressing my concerns. I have no comments.

Reviewer 3 Report

I'm satisfied with the revisions made to the article